# ON RESIDUAL NETWORK DEPTH

## ABSTRACT

Deep residual architectures, such as ResNet and the Transformer, have enabled models of unprecedented depth, yet a formal understanding of why depth is so effective remains an open question. A popular intuition, following Veit et al. (2016), is that these residual networks behave like ensembles of many shallower models. Our key finding is an explicit analytical formula that verifies this ensemble perspective, proving that increasing network depth is mathematically equivalent to expanding the size of this implicit ensemble. Furthermore, our expansion reveals a hierarchical ensemble structure in which the combinatorial growth of computation paths leads to an explosion in the output signal, explaining the historical necessity of normalization layers in training deep models. This insight offers a first-principles explanation for the historical dependence on normalization layers and sheds new light on a family of successful normalization-free techniques like SkipInit and Fixup. However, while these previous approaches infer scaling factors through optimizer analysis or a heuristic analogy to Batch Normalization, our work offers the first explanation derived directly from the network's inherent functional structure. Specifically, our Residual Expansion Theorem reveals that scaling each residual module provides a principled solution to taming the combinatorial explosion inherent to these architectures. We further show that this scaling acts as a capacity controls that also implicitly regularizes the model's complexity.

## 1 INTRODUCTION

The introduction of deep residual networks (He et al., 2016a) marked a pivotal moment in deep learning, enabling the stable training of architectures with unprecedented depth. A central intuition for their success, proposed by Veit et al. (2016), is the "unraveled view," which suggests that a ResNet functions not as a single, monolithic entity, but as an implicit ensemble of many shallower networks. This perspective, however, has remained largely conceptual, supported by empirical lesion studies rather than a precise mathematical framework.

In this work, we move beyond analogy to provide a rigorous analytical foundation for this ensemble interpretation. Our main contribution is the Residual Expansion Theorem, which derives an explicit formula for the exact functional form of the implicit ResNet ensemble. This expansion reveals a hierarchical structure where increasing network depth directly corresponds to increasing the number of models in the ensemble, which are combined in a combinatorially growing number of ways. A direct consequence of this finding is a new, first-principles explanation for the instability of very deep networks: a "combinatorial explosion" in the number of functional paths leads to an explosion in the output magnitude, a problem historically managed by normalization layers.

This theoretical insight also sheds new light on a family of successful techniques for training very deep networks without normalization. Methods like Fixup (Zhang et al., 2019) and SkipInit (De & Smith, 2020) have empirically demonstrated that scaling down residual branches is crucial for stability. Their scaling factors, however, were derived from analyzing optimizer dynamics or by analogy to Batch Normalization. Our Residual Expansion Theorem provides an underlying theoretical justification, based on the network's functional structure, explaining that these methods are effective precisely because they implicitly counteract the combinatorial explosion of the paths that we formally identify.

Our theorem not only explains why scaling is necessary but also establishes a principled framework when training residual architectures. We show that scaling each residual module by the inverse of

the total network depth (i.e., scaling by $1/n$ for a network of depth $n$) is an immediate mathematical consequence of our ensemble characterization, specifically designed to tame this combinatorial growth. However, while this $1/n$ scaling architecturally ensures stable, normalization-free training, experimentally we notice that scaling factors beyond $1/n$ and closer to $1/\sqrt{n}$, consistently lead to higher test accuracy without reintroducing the need for normalization. This suggests treating the scaling of the residual branch as an hyper-parameter $\lambda$ to be tuned around the value $1/\sqrt{n}$. In the last part of our paper, we dive into both experimental and theoretical analysis, demonstrating that $\lambda$ functions as a form of *regularized* capacity control that simultaneously enhances model capacity and promotes simpler solutions.

In summary, this paper makes the following contributions:

- We introduce the Residual Expansion Theorem, formally establishing the ensemble nature of ResNets and providing an explicit formula that links network depth to ensemble size; see Theorem 3.1.

- We pinpoint the combinatorial explosion of functional paths as the root cause of instability in deep, unnormalized residual networks, offering a unifying theoretical explanation for the efficacy of existing scaling-based methods; see Section 3.1.

- We derive a principled scaling method for residual branches, parameterized by $\lambda$, which directly counteracts the combinatorial explosion identified by our theorem, thereby enabling stable, normalization-free training of deep networks; see Section 4.1.

- We show that our proposed scaling acts as a novel form of capacity control and implicitly regularizes the model's geometric complexity; see Section 4.2 and Appendix B.

In Appendix C we provide a geometric analysis showing that the loss surfaces of shallower residual networks are naturally embedded within those of deeper ones, providing a different but complementary angle on the impact of residual depth on the optimization of residual models.

## 2 RELATED WORK

**Residual Architectures and Ensembles.**    The remarkable success of deep residual networks (ResNets), first introduced by He et al. (2016a), has prompted a significant body of research aimed at understanding the mechanisms that enable their training at extreme depths. The core architectural innovation (the identity shortcut or skip connection) was designed to address the degradation problem observed in very deep plain networks. A follow-up analysis by He et al. (2016b) further refined the architecture, arguing that unimpeded information propagation through "identity mappings" in both the forward and backward passes is crucial for stable training.

Building on this foundation, a pivotal contribution in this area is the work of Veit et al. (2016), who proposed an "unraveled view" of ResNets, interpreting them not as a single, monolithic deep model, but as an implicit ensemble of a combinatorial number of shallower networks. Each possible path that data can take through the network by either passing through a residual block or bypassing it via the identity connection constitutes a distinct member of this ensemble. The primary evidence for this interpretation came from lesion studies, which demonstrated that removing individual residual blocks at test time led to a graceful degradation in performance, akin to removing models from a conventional ensemble, whereas removing layers from a plain network caused catastrophic failure. This perspective also offered a compelling explanation for how ResNets mitigate the vanishing gradient problem: rather than preserving gradient flow through the entire network depth, gradients during backpropagation are dominated by the collection of relatively short paths, which remain trainable.

While this ensemble analogy has been highly influential, it is not exact, as the constituent paths are not independent but share parameters across layers, a key distinction from traditional ensembling methods. Our work builds directly upon this intuition by moving beyond empirical observation and analogy to provide a precise mathematical characterization. We derive an explicit formula for the exact functional form of this implicit ensemble, revealing a hierarchical structure where models of increasing complexity are combined, thereby providing an analytical foundation for the ensemble nature of residual architectures.

**Mathematical Formalisms of the Ensemble View.** Building on the conceptual foundation laid by Veit et al. (2016), other lines of research have sought to formalize the ensemble nature of residual networks through different mathematical frameworks. Huang et al. (2018) framed the ResNet architecture through the lens of boosting theory, proposing a "telescoping sum boosting" of weak learners. In their work, the final output of a ResNet is shown to be equivalent to a summation of "weak module classifiers" derived from each residual block, providing a rigorous mathematical basis for the ensemble interpretation that is analogous to the first-order term in our expansion. Another theoretical approach (Lu et al., 2020) views ResNets in a continuous limit, treating them as a discretization of an ordinary differential equation (ODE). This analysis also uncovers a combinatorial structure, showing that at order $k$, the function contains $\mathcal{O}(kL)$ terms, with the contribution of the $k$-th order term decaying as $\mathcal{O}(1/k!)$. More recently, a parallel line of work by Chen et al. (2024) introduced "Jet Expansions," a framework that also decomposes a residual network's computation into a sum of its constituent paths . This method uses jets, which generalize Taylor series, to systematically disentangle the contributions of different computational paths to a model's final prediction. While our expansion is used to derive a scaling law for stable training, the Jet Expansion framework is primarily motivated by model interpretability, enabling data-free analysis of model behavior such as extracting n-gram statistics or indexing a model's toxicity. The concurrent development of these expansion-based theories highlights a trend toward understanding network function by analyzing its underlying computational paths.

**The Role of Width and Depth.** The architectural design of neural networks involves a fundamental trade-off between depth (the number of layers) and width (the number of neurons per layer). Theoretical work has provided strong evidence for the primacy of depth in terms of expressive power. Raghu et al. (2017) demonstrated, through an analysis of activation patterns and a novel metric called "trajectory length," that the complexity of the function a network can represent grows exponentially with depth but only polynomially with width. This suggests that for a fixed parameter budget, deeper networks are theoretically capable of approximating a far richer class of functions than their shallower counterparts. However, this theoretical advantage is challenged by empirical findings. Zagoruyko & Komodakis (2016) introduced Wide Residual Networks (WRNs) and showed that a significantly wider but much shallower ResNet could outperform a very deep and thin one, while also being substantially more computationally efficient to train. They argued that extremely deep networks suffer from "diminishing feature reuse," where additional layers provide marginal benefits at a high computational cost. While depth is theoretically potent and a minimum width is necessary for universal approximation (Lu et al., 2017), the practical benefits of extreme depth have been questioned.

Our work offers a new lens through which to view this debate for residual architectures. The derived ensemble expansion reveals that increasing depth $n$ is mathematically analogous to increasing the number of experts in the model mixture. This reframes the discussion from a simple trade-off between depth and width to one between the size of the ensemble and the capacity of its individual members, offering a potential synthesis of these competing perspectives.

**Normalization Layers and Depth.** A key practical challenge in training very deep networks is maintaining stable signal propagation and avoiding the explosion of activation magnitudes. Historically, this has been addressed by normalization layers, which have become a standard component in deep learning architectures. The seminal work on Batch Normalization by Ioffe & Szegedy (2015) proposed normalizing layer inputs over a mini-batch to reduce "internal covariate shift," which dramatically stabilized training and allowed for higher learning rates. Subsequently, Layer Normalization (Ba et al., 2016) was introduced, which normalizes over the features within a single example, making it independent of batch size and particularly effective for recurrent architectures and Transformers. More recently, a line of research has explored the possibility of training deep networks without any normalization layers. Notable successes in this area include Fixup initialization Zhang et al. (2019) and SkipInit De & Smith (2020), which use careful, static rescaling of weights or branches at initialization to ensure stable dynamics. A different perspective is offered by recent work from Bordelon & Pehlevan (2025), which uses dynamical mean-field theory to analyze training in the infinite-depth limit of deep linear networks. Their analysis suggests that for the training dynamics to have a well-defined limit, residual branches must be scaled by $1/\sqrt{\text{depth}}$.

## 3 THE RESIDUAL EXPANSION THEOREM

To analyze the functional properties of deep residual networks, we consider a slightly modified but representative architecture. Most modern residual models can be described by the following general form, which separates the network into an encoding block $E_\xi$, a tower of residual blocks $R_\theta$ and a final decoding layer $D_\eta$ as follows:

$$f(x) = D_\eta \circ R_\theta \circ E_\xi(x) \tag{1}$$

where

- $E_\xi : \mathbb{R}^{d_{in}} \to \mathbb{R}^{d_e}$ is an *encoding network* that maps the input into a representational space.
- $R_\theta : \mathbb{R}^{d_e} \to \mathbb{R}^{d_e}$ is a *residual tower* composed of $n$ blocks that transforms the representation of the encoded input:

$$R_\theta = (1 + \lambda F_n) \circ \cdots \circ (1 + \lambda F_1), \tag{2}$$

  with each block having the form $(1 + \lambda F_i)(z) = z + \lambda F_i(z)$ where $F_i : \mathbb{R}^{d_e} \to \mathbb{R}^{d_e}$ is a function representing the residual branch (e.g., a sequence of linear, normalization, and activation layers). The scalar $\lambda$ controls the contribution of each residual branch, which is the slight modification we introduce.
- $D_\eta : \mathbb{R}^{d_e} \to \mathbb{R}^{d_{out}}$ is a *decoding network* that maps the final representation to the output space, which we assume to be an affine map, $D_\eta(z) = W_\eta z + b_\eta$.

This structure allows us to derive an exact expansion for the network's function, formalizing the intuition that a ResNet behaves as an ensemble of shallower networks.

**Theorem 3.1** (The Residual Expansion Theorem). *Consider a residual network with $n$ blocks of the form given in Equation 1. First of all, the residual tower admits the following expansion:*

$$R_\theta(z) = z + \lambda \sum_{i=1}^{n} F_i(z) + \lambda^2 \sum_{1 \le i < j \le n} F_j'(z) F_i(z) + \mathcal{O}(\lambda^3) \tag{3}$$

*Moreover the residual network can be expressed as a infinite sum of increasingly larger ensembles of models as a result:*

$$f(x) = \underbrace{D_\eta\big(E_\xi(x)\big)}_{\text{Order 0: Base Model } M_0} + \lambda \underbrace{\sum_{i=1}^{n} W_\eta F_i\big(E_\xi(x)\big)}_{\text{Order 1: Ensemble } M_1} + \lambda^2 \underbrace{\sum_{1 \le i < j \le n} W_\eta F_j'(E_\xi(x)) F_i(E_\xi(x))}_{\text{Order 2: Ensemble } M_2} + \dots \tag{4}$$

*Moreover, if we further assume that the encoding network is an affine map $E_\xi(x) = W_\xi x + b_\xi$, then the base model is also an affine map $D_\eta\big(E_\xi(x)\big) = W_0 x + b_0$ with $W_0 = W_\eta W_\xi$ and $b_0 = W_\eta b_\xi + b_\eta$.*

*Proof.* Let us start by showing the expansion for the residual tower as in Equation 3. We proceed by induction. For the base case, $n = 1$, this is trivial. Suppose now that this is true for any composition of $n - 1$ operators of the form $(1 + \lambda F_i(z))$. By definition, we have that

$$
\begin{aligned}
(1 + \lambda F_n) \circ \cdots \circ (1 + \lambda F_1)(z) &= (1 + \lambda F_n)\Big((1 + \lambda F_{n-1}) \cdots (1 + \lambda F_1)(z)\Big) \\
&= X + \lambda F_n(X)
\end{aligned}
$$

with $X = (1 + \lambda F_{n-1}) \cdots (1 + \lambda F_1)(z)$. Now by induction hypothesis we have that

$$X = z + \lambda \sum_{i=1}^{n-1} F_i(z) + \lambda^2 \sum_{1 \le i < j \le n-1} F_j'(z) F_i(z) + \mathcal{O}(\lambda^3)$$

Therefore, taking a Taylor series for the second term of $X + \lambda F_n(X)$, we obtain

$$\lambda F_n(X) = \lambda F_n(z) + \lambda^2 \sum_{i=1}^{n-1} F_n'(z) F_i(z) + \mathcal{O}(\lambda^3).$$

Summing up $X$ and $\lambda F_n(X)$, we obtain that the composition $(1 + \lambda F_n) \circ \cdots \circ (1 + \lambda F_1)(z)$ has the form

$$z + \lambda \sum_{i=1}^{n} F_i(z) + \lambda^2 \sum_{1 \leq i < j \leq n} F_j'(z) F_i(z) + \mathcal{O}(\lambda^3)$$

which completes the proof for ther residual tower expansion. The residual network expansion in Equation 4 follows immediately from assuming that $E_\xi(x) = W_\xi x + b_\xi$ and $D_\eta(z) = W_\eta z + b_\eta$. Namely, we have that

$$
\begin{align}
f(x) &= D_\eta\Big(R_\theta\big(E_\xi(x)\big)\Big) \tag{5}\\
&= W_\eta\Big(E_\xi(x) + \lambda \sum_{i=1}^{n} F_i(E_\xi(x)) \tag{6}\\
&\quad + \lambda^2 \sum_{1 \leq i < j \leq n} F_j'(E_\xi(x)) F_i(E_\xi(x)) + \mathcal{O}(\lambda^3)\Big) + b_\eta \tag{7}\\
&= \Big(W_\eta E_\xi(x) + b_\eta\Big) + \lambda \sum_{i=1}^{n} W_\eta F_i(E_\xi(x)) \tag{8}\\
&\quad + \lambda^2 \sum_{1 \leq i < j \leq n} W_\eta F_j'(E_\xi(x)) F_i(E_\xi(x)) + \mathcal{O}(\lambda^3) \tag{9}\\
&= D_\eta\big(E_\xi(x)\big) + \lambda \sum_{i=1}^{n} W_\eta F_i(E_\xi(x)) \tag{10}\\
&\quad + \lambda^2 \sum_{1 \leq i < j \leq n} W_\eta F_j'(E_\xi(x)) F_i(E_\xi(x)) + \mathcal{O}(\lambda^3) \tag{11}
\end{align}
$$

Moreover, if $E_\xi(x) = W_\xi x + b_\xi$ is an affine map we have that the base model is also an affine map:

$$D_\eta\big(E_\xi(x)\big) = W_\eta(W_\xi x + b_\xi) + b_\eta = W_0 x + b_0 \tag{12}$$

with $W_0 = W_\eta W_\xi$ and $b_0 = W_\eta b_\xi + b_\eta$. $\qquad\square$

### 3.1 Interpretation and the Combinatorial Explosion

The Residual Expansion Theorem provides a precise mathematical foundation for the "ensemble view" of residual networks and a first-principles explanation for their instability at extreme depths.

**Hierarchical Ensemble.** The expansion reveals a structured hierarchy of models of increasing capacity. For instance, when the encoding network is linear, then the *zero-order term*,

$$M_0(x) = W_0 x + b_0,$$

is a simple linear model with a factorized parametrization $W_0 = W_\eta W_\xi$ and $b_0 = W_\eta b_\xi + b_\eta$. This serves as the foundational model, whose loss landscape is embedded within all deeper variants (for a more detailed discussion on the geometric intuition behind this viewpoint, see Appendix C). Assuming an average scaling $\lambda = 1/n$ of the residual modules, the *first-order term*

$$M_1(x) = \frac{1}{n} \sum_{i=1}^{n} W_\eta F_i\big(E_\xi(x)\big)$$

becomes an ensemble of $n$ models, each passing through one residual block. The *second-order term*

$$M_2(x) = \frac{1}{n^2} \sum_{1 \leq i < j \leq n} W_\eta F_j'(E_\xi(x)) F_i(E_\xi(x))$$

is an ensemble of $\binom{n}{k} \simeq n^2$ more complex models made of two residual blocks. Note that in this ensemble the modules $F_j$ and $F_i$ are not simply composed as $F_j(F_i(x))$ but rather the linear approximation $F_j'(E_\xi(x))$ of $F_j$ at $E_\xi(x)$ instead is applied to $F_i(E_\xi(x))$. Beyond $M_2$ the higher-order ensembles are significantly harder to describe and interpret as they involve all higher-derivatives of the residual branches.

**Combinatorial Explosion.** The number of terms in the residual ensemble at order $k$ is bounded below by the binomial coefficient $\binom{n}{k}$. For a fixed $k$, this count grows polynomially with the depth $n$ (as $\mathcal{O}(n^k)$). Summing across all orders, the total number of terms grows exponentially as $2^n$. Without any constraints on $\lambda$, this leads to a "combinatorial explosion"; i.e., as the depth $n$ increases, the output magnitude can grow uncontrollably due to the rapidly increasing number of terms. This provides a fundamental explanation for the historical reliance on normalization layers to stabilize training in very deep residual networks. This insight directly motivates the strategy of scaling the residual branches by a factor $\lambda < 1$ to counteract this combinatorial explosion, which we explore in the next section.

## 4    THE IMPACT OF $\lambda$

In this section we discuss the effect of the scale parameter $\lambda$ on trainable depth, model capacity, and model complexity.

### 4.1    THE IMPACT OF $\lambda$ ON THE TRAINABLE DEPTH

Our Residual Expansion Theorem 3.1 shows that the number of terms at order $\lambda^n$ grows combinatorially as the number of residual layers $n$ increases. Specifically, at first order we have only $n$ terms, each of which has similar magnitude. However, already at second order in $\lambda$, the number of terms in the ensemble is of order $\mathcal{O}(n^2)$, and, more generally, the number of terms of similar magnitude at order $k$ is of order $\mathcal{O}(n^k)$. The overall combinatorial explosion of terms is exponential, and we can expect a corresponding explosion in magnitude in the model output as the number of layers increases. This fundamental issue explains why deep residual architectures have historically relied on some form of normalization layers, such as Batchnorm Ioffe & Szegedy (2015) or LayerNorm Ba et al. (2016), for stable training.

On the other hand, from Theorem 3.1, we can predict that scaling the residual branch by a factor $\lambda < 1$ will counteract the combinatorial explosion of the higher-ensemble terms at each order, provided $\lambda$ is chosen sufficiently small. Following this intuition, we should then be able to train deep residual networks without normalization but only by scaling the residual branches. As we verify in Table 1, this is indeed the case. Deeper networks become untrainable without normalization, but remain stably trainable with comparable test accuracy when their residual branches are appropriately scaled.

Our Residual Expansion Theorem 3.1 directly yields a specific choice for $\lambda$ that controls this combinatorial growth, ensuring that the magnitude of higher-order ensemble contributions remains independent of network depth. For example, if we set $\lambda = 1/n$, the magnitudes of the first two residual ensembles ($M_1$ and $M_2$) become independent of the number of layers, effectively converting the exploding sums into stable averages across multiple implicit paths. Note that, we then have

$$M_1(x) \quad = \quad \frac{1}{n} \sum_{i=1}^{n} W_\eta F_i\big(E_\xi(x)\big) \tag{13}$$

$$M_2(x) \quad = \quad \frac{1}{n^2} \sum_{1 \le i < j \le n} W_\eta F_j'(E_\xi(x)) F_i(E_\xi(x)). \tag{14}$$

Table 1 shows that it becomes difficult to train a standard residual network (i.e., $\lambda = 1$) without normalization as depth grows. However, with a simple scaling like $\lambda = 1/n$, it is possible to effectively train thousands of layers without any use of normalization layers.

**Remark 4.1.** The scaling $\lambda = 1/n$ is very natural since it makes the number of terms at each $\lambda^k$ independent of the network length $n$, essentially replacing a sum with an average. Interestingly, however, while the $1/n$ scaling factor enables stable, normalization-free training of very deep residual networks, we empirically observe a decrease in test performance compared to normalized architectures. Nevertheless, by setting $\lambda = 1/\sqrt{n}$ instead, we not only maintain stable training but also recover a substantial portion, if not all, of the performance loss attributable to the absence of normalization layers. Perhaps one reason for this is that that the average scaling $\lambda = 1/n$ may suppress the contribution of the higher-ensemble too much. Namely, if we take into account $k$ in the

estimation of the number of terms at order $\lambda^k$, we have that the number of terms grows as $\mathcal{O}(n^k/k!)$. Using $\lambda = 1/n$ is then produces an output-magnitude growth of order $\mathcal{O}(1/k!)$, which rapidly becomes small. On the contrary, setting $\lambda = 1/\sqrt{n}$ leaves a growth of order $\mathcal{O}(n^{k/2}/k!)$ which may counteract the rapid decay of the higher-order ensemble. This finding points toward a need to tune $\lambda$ or to leave it as parameter to be learned in the spirit of SkipInit (De & Smith, 2020). In the next section, we examine the impact of $\lambda$ on the learned model.

Table 1: Optimal test accuracies (with error bars) on CIFAR-10 for a deep residual model of $n$ convolutional layers. We evaluate three scaling factors for the residual branches: $\lambda \in \{1, 1/\sqrt{n}, 1/n\}$. For the $\lambda = 1$ case, results are shown with and without Batch Normalization. Note, entries are omitted where training was frozen or diverged at random initialization. See Appendix A for experiment details.

| | Number of layers ($n$) | | |
| --- | --- | --- | --- |
| | 10 | 100 | 1000 |
| $\lambda = 1.0$ with BatchNorm | $90.0 \pm 0.1$ | $89.02 \pm 0.1$ | $88.33 \pm 0.1$ |
| $\lambda = 1.0$ without BatchNorm | $85.19 \pm 0.3$ | – | – |
| $\lambda = 1/n$ without BatchNorm | $87.87 \pm 0.1$ | $87.53 \pm 0.2$ | $81.52 \pm 0.4$ |
| $\lambda = 1/\sqrt{n}$ without BatchNorm | $88.0 \pm 0.1$ | $89.02 \pm 0.09$ | $86.82 \pm 0.3$ |

## 4.2 THE IMPACT OF $\lambda$ ON CAPACITY AND COMPLEXITY

Perhaps unsurprisingly, $\lambda$ acts a capacity control forcing the model to coincide with the base model $M_0$ when $\lambda = 0$ and increasingly allowing more contribution of the higher-ensembles with higher capacity as $\lambda$ increases. This behavior as capacity control is clearly seen in our experiment of training CIFAR-10 on a simple variant of the ResNet architecture as described in Appendix A. In Figure 1, the training loss shows that the base model (for $\lambda = 0$) is underfitting with a training loss value plateauing above 1. However, increasing $\lambda$ produces training losses that are increasingly able to fully fit the data, leading to interpolation (over-parametrized regime), consistent with the interpretation of $\lambda$ as a form of capacity control.

This increase of interpolation power as $\lambda$ increases goes with a corresponding increase of test accuracy as in Figure 2 (left). In fact, *paradoxically*, the capacity increase achieved by higher-values of $\lambda$ produces models that are less complex (see Figure 2, right), as measured by the geometric complexity introduced in Dherin et al. (2022). (See Appendix B for a definition of the geometric complexity and an approximation corollary to Theorem 3.1 for residual networks). In that sense, $\lambda$ acts simultaneously as a regularizer on the model complexity and as a control on the model capacity, with higher $\lambda$ leading to better test performance *as well as to simpler models* (up to the point of divergence).

A closer analysis of the learning curves for the geometric complexity in Figure 1 shows that, in a first part of the training, higher values of $\lambda$ also come with an increase of the model complexity, as we would expect by increasing the contribution of the higher-order ensembles. However after a certain number of steps this expected behavior exactly reverses as the model enters a second stage, reminiscent of a similar phenomena observed in deep double descent as identified in Belkin et al. (2019). This reversal manifests in Figure 2 (right) as well by a decrease in geometric complexity after an initial increase as $\lambda$ goes up. We leave a precise investigation of the deeper nature of this relationship to future work.

## 5 CONCLUSION

In this work, we have introduced the Residual Expansion Theorem, a formal mathematical framework that moves the understanding of deep residual networks from the conceptual analogy of an ensemble to a precise mathematical statement. Our theorem provides an explicit formula for the network's function, revealing that its depth is mathematically equivalent to the size of a hierarchical ensemble of models. The primary insight from this expansion is the identification of a combinatorial

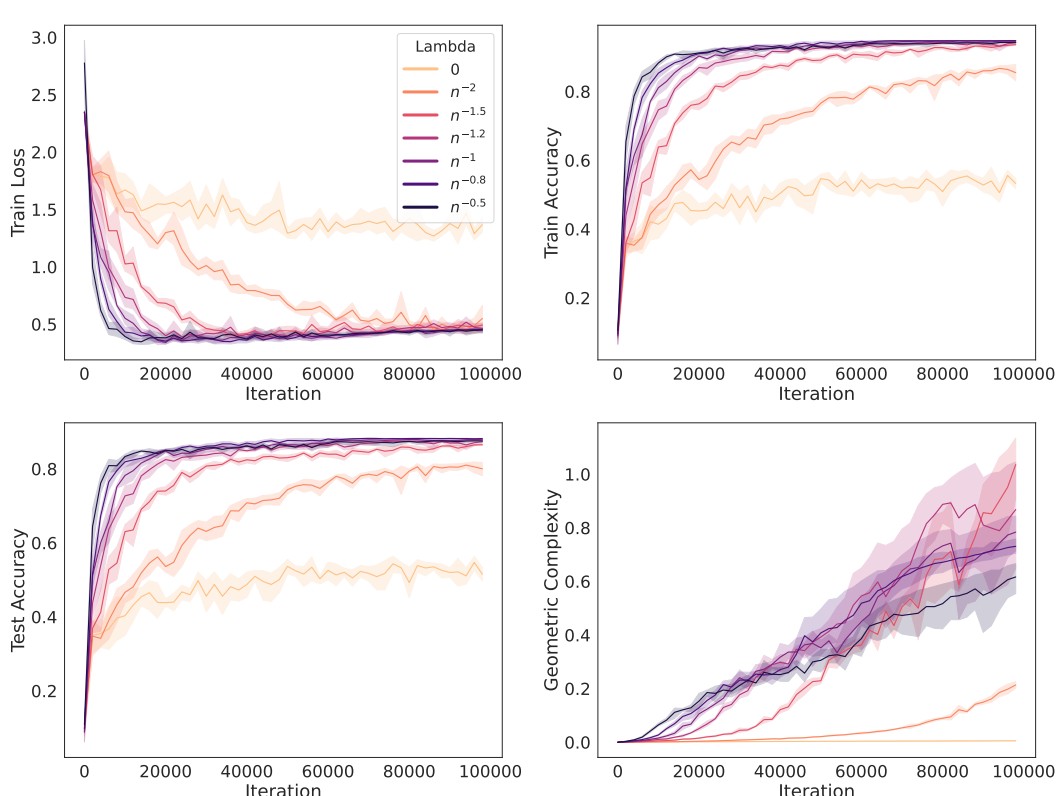

Figure 1: CIFAR-10 trained on residual network with $n = 16$ residual blocks. We plot the learning curves for the experiments in Figure 2 for a sweep $\lambda \in \{0, n^{-2}, n^{-1.5}, n^{-1.2}, n, n^{-0.8}, n^{-0.5}\}$. Training with $\lambda > 1/\sqrt{n}$ (e.g., we also tried $\lambda \in \{1/n^{0.3}, 1/n^{0.4}, 1\}$) all failed.

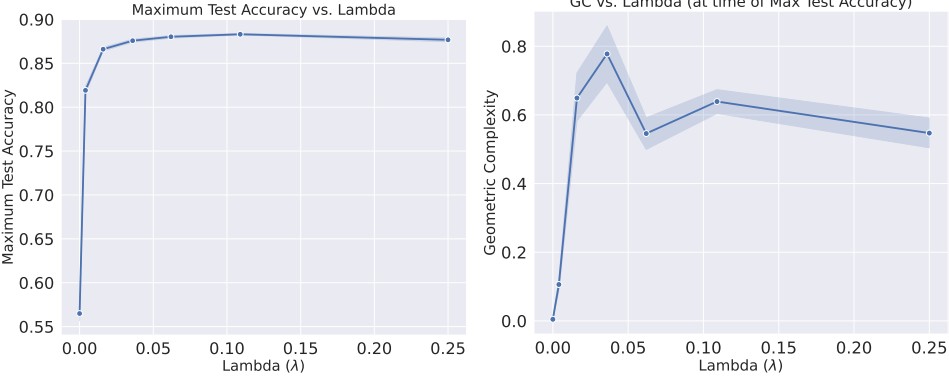

Figure 2: Maximum test accuracy and geometric complexity at time of maximum test accuracy for various values of $\lambda$. **Left:** As $\lambda$ increases, maximum test accuracy increases. **Right:** However, increasing $\lambda$ leads to decreased model complexity after a first phase of increase.

explosion of functional paths, which we posit as the fundamental cause of the training instability that has historically necessitated the use of normalization layers.

This theoretical lens provides a new, unifying explanation for the success of a family of existing normalization-free training methods. Techniques such as Fixup and SkipInit have empirically demonstrated the need for scaling down residual branches to achieve stability. However, their scaling factors were derived from analyzing optimizer dynamics or by analogy to Batch Normalization. Our work provides the first justification grounded in the network's functional structure, showing that these methods are effective precisely *because* they serve as practical implementations of a necessary principle: taming the combinatorial growth of paths.

This work opens several promising avenues for future research. The contrast between our finite-depth $1/n$ scaling and the $1/\sqrt{n}$ scaling derived from infinite-depth mean-field theories (for deep linear networks Bordelon & Pehlevan (2025)) suggests a rich spectrum of principled scaling laws that warrant further investigation. Applying this function-first scaling paradigm to other critical architectures, particularly Transformers, stands as a crucial next step. Ultimately, by providing a unifying theoretical foundation, this work paves the way for a more principled design of robust and extremely deep neural networks.

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

## A  EXPERIMENT DETAILS

### A.1  RESNET ARCHITECTURE DETAILS

For the values reported in Table 1 and the plots in Figures 1 and 2, we train a deep residual network consisting of $n$ convolutional blocks. Similar to a traditional ResNet model He et al. (2016a), each layer consists of two convolutions with $(3, 3)$ kernels and ReLU activations. Our architecture differs from the standard ResNet in that we keep the number of channels fixed at 256 and we only use strides of size $(1, 1)$. These two changes ensure that the shape remains fixed as information flows through the network between layers. Also, for our model to be consistent with the theory, we remove the final ReLU in each residual block. Therefore, a single version of our residual block takes the form

$$\text{ResNetBlock}(x) = x + \lambda F(x), \qquad \text{where } F(x) = \text{Conv}_{3\times3}(\text{ReLU}(\text{Conv}_{3\times3}(x)))$$

Our residual tower consists of stack of $n$ of these Residual Blocks. For those models trained with Batch Normalization, we modify the transformation $F$ so that

$$F(x) = \text{BatchNorm}(\text{Conv}_{3\times3}(\text{ReLU}(\text{BatchNorm}(\text{Conv}_{3\times3}(x))))).$$

This describes the residual tower $R_\theta$ as discussed in Section 3. For the encoding $E_\xi$ and decoding $D_\eta$ networks, we use the same architecture as that of a traditional ResNet model as implemented in Heek et al. (see for instance `https://github.com/google/flax/blob/main/examples/imagenet/models.py`).

### A.2  EXPERIMENT DETAILS FOR TABLE 1

All results in Table 1 are obtained by training the CIFAR-10 dataset Krizhevsky (2009) on the residual network described above. The only exception is for models of depth $n = 1000$, where the architecture's width (or channel size) was set to 128 instead of 256 because of memory limitation; all other aspects remained identical. Each model was trained with a batch size of 64 for 160,000 steps (i.e., approximately 200 epochs) using SGD with momentum 0.9 and on a sweep of learning rates ranging from $\{2^{-10}, 2^{-9}, \ldots, 2^{-1}\}$. We apply data augmentation in the form of data scaling, random cropping and random flips. We do not employ any learning rate schedule or weight decay during training.

For each learning rate in the sweep, we train the model with 5 random seeds which are used for model initialization and data shuffling. We then measure and report the mean and standard error for the best average test accuracy achieved for a given learning rate.

### A.3  EXPERIMENT DETAILS FOR FIGURE 1 AND FIGURE 2

All results in Figure 1 (and Figure 2) are obtained by training the CIFAR-10 dataset Krizhevsky (2009) on the residual network described in Section A.1. The number of residual blocks was set to $n = 16$. Each model was trained with a batch size of 64 for 100,000 steps (i.e., approximately 80 epochs) using SGD with momentum 0.9 and learning rate 0.3. We apply data augmentation in the form of data scaling, random cropping and random flips. We do not employ any learning rate schedule or weight decay during training.

For each $\lambda \in \{0, n^{-2}, n^{-1.5}, n^{-1.2}, n, n^{-0.8}, n^{-0.5}, n^{-0.4}, n^{-0.3}, 1\}$ in the sweep (with $n = 16$), we train the model with 5 random seeds which are used for model initialization and data shuffling. All the values of $\lambda$ above $1/\sqrt{n}$ diverged in that setting.

## B  GEOMETRIC COMPLEXITY

The *geometric complexity* (Dherin et al., 2022) is a measure of how much a model's output changes in response to small changes in its input. Models with high geometric complexity can represent intricate, jagged functions, which are prone to overfitting, whereas models with lower complexity are constrained to learn smoother, simpler functions that often generalize better.

More precisely, the geometric complexity of a model $f$ over a dataset $D$ is defined as the average squared Frobenius norm of its input-output Jacobian:

$$\langle f, D \rangle_G := \frac{1}{|D|} \sum_{x \in D} \|\nabla_x f(x)\|_F^2$$

The Residual Expansion Theorem allows us to approximate how the geometric complexity changes with $\lambda$. In fact, the following corollary shows that for a residual models as in Equation 1, the model geometric complexity corresponds to the geometric complexity of the base model $M_0$ plus a term depending on $\lambda$. The fact that this latter term can be negative (as it is an inner product in matrix space) shows the theoretical possibility that models with higher $\lambda$ can be geometrically less complex than model with smaller $\lambda$ (see Figure 2, right).

**Corollary B.1.** *The geometric complexity of a residual network as defined in Equation 1 is given at first orders by:*

$$\langle f, D \rangle_G = \langle M_0, D \rangle_G + 2\lambda \frac{1}{|D|} \sum_{x \in D} Tr\left( \left(W_\eta E'_\xi(x)\right)^T \sum_{i=1}^n W_\eta F'_i(E_\xi(x)) E'_\xi(x) \right) + \mathcal{O}(\lambda^2)$$

*Proof.* To evaluate the geometric complexity of $f$ over the dataset $D$, we need to evaluate the derivative of $f$. We can write it as

$$\nabla_x f(x) = A(x) + \lambda B(x) + \mathcal{O}(\lambda^3), \tag{15}$$

where

$$A(x) = \nabla_x \left( D_\eta \circ E_\xi \right)(x) = W_\eta E'_\xi(x) \tag{16}$$

$$B(x) = \nabla_x \left( \sum_{i=1}^n W_\eta F_i\left(E_\xi(x)\right) \right) = \sum_{i=1}^n W_\eta F'_i\left(E_\xi(x)\right) E'_\xi(x). \tag{17}$$

Now the Frobenius norm of the model derivative can be written as

$$\|\nabla_x f(x)\|_F^2 = Tr\left( f'(x)^T f(x) \right) \tag{18}$$

$$= Tr\left( \left(A^T + \lambda B^T + \mathcal{O}(\lambda^2)\right)\left(A + \lambda B + \mathcal{O}(\lambda^2)\right) \right) \tag{19}$$

$$= Tr(A^T A) + 2\lambda Tr(A^T B) + \mathcal{O}(\lambda^2) \tag{20}$$

$$= \|\nabla_x \left( D_\eta \circ E_\xi(x) \right)\|_F^2 \tag{21}$$

$$+ 2\lambda Tr\left( \left(W_\eta E'_\xi(x)\right)^T \sum_{i=1}^n W_\eta F'_i(E_\xi(x)) E'_\xi(x) \right) + \mathcal{O}(\lambda^2), \tag{22}$$

where we used that $Tr(B^T A) = Tr(A^T B)$. We now obtain the desired result by averaging $\|\nabla_x f(x)\|_F^2$ over the data $D$. $\square$

## C  THE GEOMETRIC COUNTERPART TO THE RESIDUAL EXPANSION

In this section, we provide the geometric intuition that underpins the functional analysis presented in the main paper. We show how the structure of the parameter space and the corresponding loss landscape provides a complementary perspective on why deep residual networks are effective and how our proposed scaling ensures their stability. The analysis in this section stems from the same fundamental property that enables the Residual Expansion Theorem: a residual block, $(1 + \lambda F_{\theta_i})(z)$,

becomes an identity map when its parameters are set to zero (i.e., $F_{\theta_i=0}(z) = 0$). While the main paper uses this property to derive a functional expansion, here we explore its profound implications for the geometry of the optimization problem as network depth increases. First, let us establish our setting. For a fixed dataset $\mathcal{D}$, we consider a loss function $\mathcal{L}(y, f_\theta(x))$ that depends only on the network's prediction. The total loss is the average over the dataset:

$$\mathcal{L}(\theta) = \frac{1}{|\mathcal{D}|} \sum_{(x,y)\in\mathcal{D}} \mathcal{L}(y, f_\theta(x))$$

Crucially, this means that if two networks with different parameters, $\theta$ and $\eta$, compute the same function ($f_\theta(x) = g_\eta(x)$ for all $x$), their losses are identical ($\mathcal{L}(\theta) = \mathcal{L}(\eta)$). We also assume the loss is zero for any parameter set that perfectly interpolates the data. A residual network of depth $n$, denoted $f^n_{\omega_n}(x)$, has parameters $\omega_n = (\eta, \xi, \theta_1, ..., \theta_n)$ in a parameter space $\Omega_n$. Because any residual block becomes an identity map when its parameters $\theta_i$ are zero, a network of depth $n$ contains all shallower networks of depth $k < n$ within its parameter space. For instance, by setting the parameters of the final block to zero ($\theta_n = 0$), the $n$-layer network becomes functionally identical to an $(n-1)$-layer network:

$$f^n_{(\eta,\xi,\theta_1,...,\theta_{n-1},0)}(x) = f^{n-1}_{(\eta,\xi,\theta_1,...,\theta_{n-1})}(x)$$

This structural embedding has a direct consequence on the loss landscape. The loss of the deeper network on this embedded subspace is identical to the loss of the shallower network:

$$\mathcal{L}_n(\eta, \xi, \theta_1, ..., \theta_{n-1}, 0) = \mathcal{L}_{n-1}(\eta, \xi, \theta_1, ..., \theta_{n-1})$$

This provides a powerful geometric explanation for why increasing depth is an effective optimization strategy. Adding a new layer does not force the optimizer to find a solution in an entirely new landscape; instead, it expands the search space while preserving all solutions found by shallower networks. This is particularly true for the set of global minima (the zeros of the loss function), denoted $\mathcal{Z}_n$. If a parameter set $\omega_{n-1}$ is a global minimum for the $(n-1)$-layer network (i.e., $\omega_{n-1} \in \mathcal{Z}_{n-1}$), then the point $\omega_n = (\omega_{n-1}, 0)$ must also be a global minimum for the $n$-layer network. This guarantees a natural embedding of the sets of optimal solutions:

$$\mathcal{Z}_{n-1} \hookrightarrow \mathcal{Z}_n$$

This hierarchical structure of the loss landscape is the geometric counterpart to the Residual Expansion Theorem. The zero-order term of our expansion, $P_\eta \circ E_\xi(x)$, corresponds to the base shallow network whose optimal solutions are preserved and embedded within all deeper networks. This connection also clarifies the crucial role of the $1/n$ scaling proposed in the main paper. The embedded geometry guarantees that as we add layers, "good" solutions from shallower networks continue to exist. However, the Residual Expansion Theorem shows that without proper scaling, the landscape around these solutions becomes increasingly unstable due to the combinatorial explosion of higher-order terms. The $1/n$ scaling tames this explosion, effectively smoothing the loss landscape and ensuring that the theoretically guaranteed optimal regions remain practically accessible to the optimizer, even at extreme depths.

## D   DISCLOSURE OF LLM USAGE

We used Gemini 2.5 Pro to help us with the editing and polishing of this paper. Our approach was to first write paragraphs and sections manually and ask Gemini for a clearer rewrite. We then merge both versions manually. We also used Gemini to help us to identify similarities between the results in this paper to other papers in the literature, as a form of discovery engine and summarizing tool.

