# OpenReview forum: "On residual network depth"
_ICLR.cc/2026/Conference — ICLR 2026 Conference Withdrawn Submission_

### Official Review · Reviewer_2E8g · 2025-10-22

**Soundness:** 3
**Presentation:** 3
**Contribution:** 1
**Rating:** 2
**Confidence:** 4

**Summary:**

The paper investigates the effect of depth of residual architectures on their functional behavior. The authors theoretically support the hypothesis that deep resnets effectively act as an ensemble of shallower modules. To do this, they prove the Residual Expansion Theorem – a formula for resnet’s functional behavior based on Taylor expansion and binomial expansion, which turns the composition of functions into sum of products of much shallower components. This formula shows that ResNet acts as an ensemble of hierarchically ever-more-complex ensembles. From the formula, the authors then derive a theoretically sound scaling of the residual components, which enables training on extreme depths (1000 layers) without normalization layers. They experiment with this scaling ($1/n$ where n is the number of layers) and compare it with constant scaling and $1/\sqrt{n}$ (as well as some other scalings). They show that already the $1/\sqrt{n}$ scaling suffices for a stable training and can reach higher test accuracy. They attribute this to the higher-order terms in the expansion that get less suppressed by this bigger scaling. Finally, the authors analyze the geometric complexity of learned features and its evolution throughout the training and show that it exhibits a unimodal behavior as a function of residual connection scaling, meaning that large scaling, though promoting complexity, also probably acts as an effective regularizer, because it leads to smaller geometric complexity.

**Strengths:**

-	S1: The paper studies very interesting and important topic of the behavior of ResNets in terms of their depth. The paper analyzes the hypothesis that the resnet can be seen as an ensemble of shallower modules (all the way down to individual resnet layers) of hierarchically increasing complexity. Deciding, whether such a hypothesis is true would move us a long way forward in our understanding of the effect of residual connection on the behavior of deep neural networks.
-	S2: The paper is clearly written and easy to follow.
-	S3: The expansion of residual layers into sum of products using Taylor expansion (and thus turning the compositions into simpler operations) is neat and can be of independent interest.

**Weaknesses:**

-	W1 (major): The residual expansion theorem, although technically correct, is meaningless in practice. This is because with the scaling of $\lambda \propto n^{-1}$ that the authors suggest, but also with the larger scaling of $\lambda \propto n^{-1/2},$ the term which authors call $\mathcal{O}(\lambda^3)$ is not negligible and therefore the Taylor approximation up to the second order does not say anything about the real behavior of the resnet.
-	W2 (major): Except for the Taylor expansion and experiments with geometric complexity, the entire discussion of the paper as well as Theorem 3.1 is basically just an anecdotal explanation of the binomial expansion of $(1+x)^n$ and of the fact that the scaling of $x \propto n^{-1}$ that makes the limit non-trivial. Moreover, as shown, for instance, in [1], that this scaling is only appropriate with either deterministic or at least correlated initialization regime, while for IID initialization, the scaling $x \propto n^{-1/2}$ leads to non-trivial limit, which also explains why the authors observe that this scaling still leads to stable behavior.
-	W3 (major): the authors seem to critically undercite the related work, especially around the proper initialization and the proper scaling regimes and their continuous-time limits (neural ODEs or SDEs). See for instance [1] and all the references therein. There seems to be quite vast literature that analyzes resnets from a similar point of view than the paper but the authors only cite very few of those works.
-	W4: the  amount of content of the paper is fairly modest. There is only one Theorem that only extends the binomial formula in the use of Taylor expansion. Then there are scaling experiments very similar to those already done in related work (see again [1] and refernces therein). Only the geometric complexity experiments seem to be fully novel but insufficiently discussed.
-	W5: Many resnets use ReLU as the activation function. Taylor expansion cannot be used around non-differentiable points and thus the Resiudal Expansion Theorem doesn’t apply. The authors should discuss this limitation.

[1] Marion, Pierre, et al. "Scaling ResNets in the large-depth regime." Journal of Machine Learning Research 26.56 (2025): 1-48.

**Questions:**

-	Q1: Did you try to empirically measure the relative sizes of the contributions of the ever-more-complex ensembles at initialization and at convergence? Are the single and double terms really dominating? Are the higher-order terms negligible?

---

### Official Review · Reviewer_a9GC · 2025-10-24

**Soundness:** 3
**Presentation:** 3
**Contribution:** 2
**Rating:** 2
**Confidence:** 4

**Summary:**

Research since the introduction of ResNets has explored the idea that a deep ResNet behaves like an implicit ensemble of many shallower networks, rather than one monolithic model. The seminal work of Veit et al. (2016) first articulated this “unraveled view.” They showed empirically that a ResNet can be seen as a collection of numerous paths of varying lengths through the network. While researchers had already zeroed in on the idea that residual connections must be down-scaled as depth increases to ensure training is tractable. The Residual Expansion Theorem, nicely unifies those earlier insights under one theoretical umbrella.

**Strengths:**

The work formally confirms that ResNet depth = exponentially many shallow models, and that scaling residuals = essential regularizer to tame that exponential. In doing so, it bridges gaps between these earlier results and provides a common explanation backed by theorem

**Weaknesses:**

While the paper introduces a formal expansion to characterize the ensemble structure of residual networks, several aspects of the contribution are overstated or insufficiently supported:

The hierarchical decomposition of a ResNet into subnetworks defined by which residual blocks are active (e.g., n subnetworks with a single active block, n choose 2 with two active blocks, etc.) is well-known and has been articulated in prior work, most notably by Veit et al. (2016). The observation that such subnetworks contribute terms of order λ, λ², and so on, follows directly from binomial expansion and does not require the technical machinery presented in this paper. While the authors offer a formal statement (the Residual Expansion Theorem), the insights it provides are largely intuitive and do not significantly advance understanding beyond prior conceptual or first-order analyses.

The treatment of the scaling factor λ lacks both novelty and rigor. Although the authors propose λ = 1/n as a principled scaling rule to mitigate the growth of higher-order terms, the motivation is mostly heuristic. The argument relies on treating the expansion coefficients as averages (or "natural" or "simple"), without a clear theoretical justification for why this scaling is optimal or necessary. Moreover, the authors themselves acknowledge that λ = 1/√n empirically performs better, and that tuning λ as a hyperparameter can further improve results. This undermines the claim that the paper offers a theoretically grounded or practically useful prescription for normalization-free training.

The paper's central claim that deep ResNets suffer from a “combinatorial explosion” in the number of computational paths, and that this is the root cause of instability, remains unconvincing. While it is true that the number of terms grows exponentially with depth, the paper does not clearly demonstrate how this affects actual training dynamics. For example, there is no formal connection between path count and exploding activations, gradients, or loss values in practice. Without such analysis or compelling empirical evidence, the combinatorial explosion remains a descriptive rather than explanatory concept.

**Questions:**

see weaknesses

---

### Official Review · Reviewer_xdXM · 2025-10-30

**Soundness:** 3
**Presentation:** 2
**Contribution:** 1
**Rating:** 2
**Confidence:** 4

**Summary:**

The paper studies a general formulation for residual networks of the form $X + \lambda F(X)$, where $F$ is the residual branch and $\lambda$ is a scalar controlling its contribution relative to the skip connection. The main result is the residual expansion theorem, which associates the residual network with a particular ensemble. The paper then analyzes the role of $\lambda$ and how it affects the trainability of the network.

**Strengths:**

- The paper provides a clean and precise formulation of the ensemble induced by residual networks, consisting of sums of residual branches and their derivatives.

- The theoretical analysis appears sound, is supported by experiments, and the exposition is clear and well-written.

**Weaknesses:**

**Weaknesses**

- The connection between the residual scaling $\lambda$ and network trainability has been discussed extensively in prior work (e.g., [1,2,3,4]), including the difference between $1/\text{depth}$ and $1/\sqrt{\text{depth}}$ scalings. It is also known that without appropriate residual scaling the activation norms diverge as depth increases. Given this context, it is unclear what *new* insights this work contributes to our understanding of trainability.

- The residual expansion theorem is technically sound, but relies on standard tools (e.g., Taylor expansion in $\lambda$, unrolling the recursion). In its current form, the result is mathematically correct but does not yet deliver strong conceptual intuition beyond existing literature.


**Suggestions.**
To increase the impact, I suggest extending the study to ensembles generated by concrete neural network architectures (e.g., attention layers, fully-connected networks) under realistic initializations such as Gaussian + He initialization. Demonstrating how the ensemble behaves in these practical settings would help giving more depth to the contributions.


Overall, I feel in the current form the paper significantly lacks of sufficient contributions to be in the acceptance range.


[1] Hayou et al, Stable Resnet

[2] Bordelon et al, Depthwise Hyperparameter Transfer in Residual Networks: Dynamics and Scaling Limit

[3] Yang et al, Tensor Programs VI: Feature Learning in Infinite-Depth Neural Networks

[4] Dey et al, Don't be lazy: CompleteP enables compute-efficient deep transformers

**Questions:**

See weaknesses

---

### Official Review · Reviewer_3DoG · 2025-11-01

**Soundness:** 2
**Presentation:** 2
**Contribution:** 2
**Rating:** 2
**Confidence:** 4

**Summary:**

This paper proposes a residual expansion theorem which explains how residual networks work as an ensemble of several models. Using the theorem, a scaling strategy is proposed to train residual networks without normalization. In the scaling strategy, the output of each residual block is multiplied with a parameter $\lambda$ where $\lambda<1$. Several experiments are conducted with CIFAR-10, and it is shown that the scaling factor acts as a capacity control where a larger scaling factor gives a higher ensemble and thus better performance.

**Strengths:**

The paper is written with good clarity, and the topic is important to advancing our theoretical understanding of residual networks.

**Weaknesses:**

There are several theoretical papers studying residual networks, but some important related works are missing. For example, the papers on how residual networks work, how they outperform best linear predictors, and how they can be viewed as basis function models. See the following references for example.

The residual expansion theorem seems trivial and restricted. The coefficients applied to residual blocks can be more generalized. We can have different $\lambda_1$, $\lambda_2$, $\cdots$, $\lambda_n$ applied to the residual blocks.

Another important feature of residual networks is that a residual block in a ResNet uses the output of the previous block. However, this relation is not being studied in the paper. The models in an ensemble are not entirely independent, and this may have some interesting implications.

The accuracies in the CIFAR-10 experiments seem to be much lower than the accuracies achieved by ResNets (>92% accuracy). It is difficult to see how this scaling strategy can

[1] Hardt, Moritz, and Tengyu Ma. "Identity Matters in Deep Learning." In International Conference on Learning Representations. 2017.

[2] Yun, Chulhee, Suvrit Sra, and Ali Jadbabaie. "Are deep ResNets provably better than linear predictors?." Advances in Neural Information Processing Systems 32 (2019).

[3] Chen, Kuan-Lin, Ching-Hua Lee, Harinath Garudadri, and Bhaskar D. Rao. "ResNEsts and DenseNEsts: Block-based DNN models with improved representation guarantees." Advances in neural information processing systems 34 (2021).

**Questions:**

1.	Why are the accuracies (with batch normalization) in Table 1 lower than normal CIFAR-10 results using ResNets?
2.	Dose Table 1 imply that the scaling strategy results in inferior performance? If yes, what is the practical usage of it?
3.	The paper mentions Transformer. Can this scaling strategy be applied to Transformer or other architectures? It seems to be unique to residual networks.
4.	Can we use a different $\lambda$ in every block and make them trainable?
5.	Is there a proof on why $\frac{1}{\sqrt{n}}$ gives the best results?

---

### Note · Authors · 2025-11-13

I have read and agree with the venue's withdrawal policy on behalf of myself and my co-authors.